# Sample Preparation Approach Influences PAM50 Risk of Recurrence Score in Early Breast Cancer

**DOI:** 10.3390/cancers13236118

**Published:** 2021-12-04

**Authors:** Tonje G. Lien, Hege Oma Ohnstad, Ole Christian Lingjærde, Johan Vallon-Christersson, Marit Aaserud, My Anh Tu Sveli, Åke Borg, on behalf of OSBREAC, Øystein Garred, Elin Borgen, Bjørn Naume, Hege Russnes, Therese Sørlie

**Affiliations:** 1Department of Cancer Genetics, Institute for Cancer Research, Oslo University Hospital, P.O. Box 4953 Nydalen, N-0424 Oslo, Norway; Tonje.Lien@rr-research.no (T.G.L.); ole@ifi.uio.no (O.C.L.); h.e.g.russnes@ous-research.no (H.R.); 2Department of Oncology, Division of Cancer Medicine, Oslo University Hospital, P.O. Box 4953 Nydalen, N-0424 Oslo, Norway; HEGEO@ous-hf.no (H.O.O.); bjorn.naume@medisin.uio.no (B.N.); 3Centre for Bioinformatics, Department of Informatics, University of Oslo, Gaustadalléen 23 B, N-0373 Oslo, Norway; 4Division of Oncology, Department of Clinical Sciences Lund, Lund University, Medicon Village, SE-22381 Lund, Sweden; johan.vallon-christersson@med.lu.se (J.V.-C.); ake.borg@med.lu.se (Å.B.); 5Department of Pathology, Oslo University Hospital, P.O. Box 4953 Nydalen, N-0424 Oslo, Norway; mhelg@ous-hf.no (M.A.); myahtu@ous-hf.no (M.A.T.S.); UXYSGA@ous-hf.no (Ø.G.); EBG@ous-hf.no (E.B.); 6Institute of Clinical Medicine, Faculty of Medicine, University of Oslo, P.O. Box 1171 Blindern, N-0318 Oslo, Norway

**Keywords:** Prosigna, risk of recurrence score, PAM50, FFPE, macrodissection, bulk, fresh-frozen, breast cancer

## Abstract

**Simple Summary:**

The PAM50 risk of recurrence (ROR) score is predictive of the risk of distant recurrence and the benefits of adjuvant therapy in early breast cancer. The Prosigna assay utilizes RNA from tumor cells selected via macrodissection of formalin-fixed, paraffin-embedded (FFPE) tissue sections to measure the activity of the PAM50 genes. An alternative and widely used extraction method is RNA purification from fresh-frozen (FF) bulk tissue without enriching the tumor cellularity. However, the impact of the RNA preparation approach on ROR scores and subsequent treatment selection has not been systematically evaluated. We compared the different approaches and found high correlation between risk of recurrence scores estimated from macrodissected FFPE tissue and scores estimated from bulk FF tumor tissue. However, important discrepancies were revealed for luminal tumors, which may have consequences for treatment recommendations for these patients.

**Abstract:**

The PAM50 gene expression subtypes and the associated risk of recurrence (ROR) score are used to predict the risk of recurrence and the benefits of adjuvant therapy in early-stage breast cancer. The Prosigna assay includes the PAM50 subtypes along with their clinicopathological features, and is approved for treatment recommendations for adjuvant hormonal therapy and chemotherapy in hormone-receptor-positive early breast cancer. The Prosigna test utilizes RNA extracted from macrodissected tumor cells obtained from formalin-fixed, paraffin-embedded (FFPE) tissue sections. However, RNA extracted from fresh-frozen (FF) bulk tissue without macrodissection is widely used for research purposes, and yields high-quality RNA for downstream analyses. To investigate the impact of the sample preparation approach on ROR scores, we analyzed 94 breast carcinomas included in an observational study that had available gene expression data from macrodissected FFPE tissue and FF bulk tumor tissue, along with the clinically approved Prosigna scores for the node-negative, hormone-receptor-positive, HER2-negative cases (n = 54). ROR scores were calculated in R; the resulting two sets of scores from FFPE and FF samples were compared, and treatment recommendations were evaluated. Overall, ROR scores calculated based on the macrodissected FFPE tissue were consistent with the Prosigna scores. However, analyses from bulk tissue yielded a higher proportion of cases classified as normal-like; these were samples with relatively low tumor cellularity, leading to lower ROR scores. When comparing ROR scores (low, intermediate, and high), discordant cases between the two preparation approaches were revealed among the luminal tumors; the recommended treatment would have changed in a minority of cases.

## 1. Introduction

Molecular classification of breast cancer as a basis for more precise treatment is increasingly used in clinical practice. In early breast cancer, the FDA-approved and CE-marked Prosigna assay provides a risk of recurrence (ROR) score that predicts the prognosis for patients with hormone-receptor-positive, HER2-negative tumors. This score refines patient stratification for improved treatment decisions, leading to alleviation of chemotherapy for some patients [1,2,3,4]. Prosigna uses gene expression measurements of a panel of 50 genes (PAM50) [5,6] obtained from RNA extracted from formalin-fixed, paraffin-embedded (FFPE) tumor tissue. 

Collection of FFPE tissue for molecular analysis is the standard procedure in clinical routine diagnostics. Even though the RNA often is highly fragmented, it has been shown for targeted gene panels that the data obtained from FFPE specimens are comparable with those obtained from fresh-frozen (FF) tissue [7]. However, with the increasing number of biomarkers and gene signatures in clinical use, high RNA quality for next-generation sequencing (NGS) purposes is warranted [8], but which can be difficult to obtain from FFPE tissue. For whole-exome sequencing, fresh-frozen tumor tissue is preferred.

Another critical aspect in molecular cancer diagnostics is tumor cellularity and the possible influence of normal cell infiltration. In NanoString’s pipeline for the Prosigna assay, macrodissection of the tumor area is performed on each preselected FFPE tissue section. A trained pathologist outlines the tumor area, avoiding pre-invasive foci and non-tumor tissue, before the macrodissected tumor tissue is collected and used for RNA extraction. In the case of extracting RNA from FF tissue, most protocols entail small biopsies obtained prior to routine pathological assessment, and without enrichment of tumor cells. It is therefore of interest to compare ROR scores obtained from FFPE vs. FF specimens with respect to tumor cellularity, and to evaluate their potential impact on treatment decisions. 

The ROR score takes into consideration the correlation to four of the five molecular intrinsic subtypes: basal-like, HER2-enriched, luminal A, and luminal B. The fifth subtype—the normal-like—is characterized by low expression of proliferation-associated genes, along with a gene expression pattern reminiscent of normal epithelial cells [6,9]; it is still unclear as to what extent this represents a real tumor subtype, or merely the degree of infiltrating non-tumor cells.

In this study, we analyzed tumor tissue from 94 early breast cancer patients, and compared the ROR scores obtained from macrodissected FFPE tissue with the ROR scores obtained from fresh-frozen bulk tissue. In addition, the approved Prosigna scores were available for all patients, yielding a unique dataset for comparing differences in risk assessments. Overall, the concordance between scores from FFPE and FF tissue was high; however, higher normal cell infiltration in the fresh-frozen, non-macrodissected tissue may have implications for the ROR score.

## 2. Materials and Methods

### 2.1. Ethics Statement

Ethical approval (number 29668) was obtained from the Norwegian Regional Committee for Medical Research Ethics. Before participating in the study, we obtained a written informed consent form from all participants to collect and study molecular and clinical data.

### 2.2. Patient and Tumor Characteristics

Tumors from 94 patients were obtained from the Oslo2 breast cancer cohort [10], which was an observational study collecting material from breast cancer patients with primary operable disease from several hospitals in South-Eastern Norway between 2007 and 2019. Patients included in this study were diagnosed in 2015 and 2016. An overview of clinical characteristics is given in Table 1. A complete overview of the clinical annotation for all 94 patients is found in Appendix A.

### 2.3. RNA Extraction and Gene Expression Analysis from FFPE Tissue

From FFPE tissue, RNA purification was performed using the Roche^®^ FFPET RNA Isolation Kit, Roche-025 (NanoString Technologies, Seattle, WA, USA): 1–6 FFPE slides depending on the tumor area, (4–19 mm2: 6 slides, 20–99 mm2: 3 slides, and ≥100 mm2: 1 slide). The recommended RNA input was 250 ng for the hybridizations on the nCounter Analysis System. Two assays were employed: first the Prosigna assay (reporting the final ROR and subtype of each sample, i.e., not the correlation to the subtypes), and second the PAM50 assay (reporting the raw gene expression counts for the 50 breast-cancer-related genes). The raw counts were then normalized using all housekeeping genes and, finally, log2-transformed. The normalized RNA-Seq data from the PAM50 genes can be found in Appendix A.

### 2.4. RNA Extraction and Gene Expression Analysis from FF Tissue

RNA extraction from FF tissue and subsequent sequencing using Illumina NextSeq500 has previously been described in detail [11]. The raw counts were processed as described in [12], and the resulting FPKM (fragments per kilobase per million) gene expression values were log2-transformed prior to selecting the PAM50 genes for further analysis. Tumor vs. normal cell content in all FF tissue specimens used for RNA extraction was evaluated by a pathologist. The normalized log 2-transformed nCounter counts for the PAM50 genes can be found in Appendix A.

### 2.5. Gene Centering and Subtype Classification

On gene expression data, we performed subtype classification as follows, separately for the FFPE and FF datasets: Expression values were first centered across all tumors for each gene. To account for the relatively higher number of ER-positive patients in our cohort (~80%) compared to the training dataset from Parker et al. [5] (60%), we first found the mean expression value for each gene among ER-positive and ER-negative tumors separately. The bimodal distribution of ERS1 gene expression was used to determine the ER status; then, a weighted average of the two means was found, with 60% weight on the ER-positive mean and 40% on the ER-negative mean. From the gene-centered dataset, we calculated the Pearson’s correlation of the expression vector to each of the five subtype centroids (including the normal-like centroid) for each sample, as described by Parker et al. We assigned a sample to the subtype with the highest Pearson’s correlation coefficient; all cases were given a subtype. For the Prosigna assay, it has previously been shown that only 46 of the original 50 genes are needed, excluding the genes *BIRC5, CCNB1, GRB7,* and *MYBL2* [1]; we therefore also excluded these genes in the above subtype classification. 

### 2.6. Proliferation Score

The proliferation score was calculated according to Wallden et al. [2], using the arithmetic mean of the expression of 18 proliferation genes: *ANLN, CEP55, ORC6L, CCNE1, EXO1, PTTG1, CDC20, KIF2C, RRM2, CDC6, KNTC2, TYMS, CDCA1, MELK, UBE2C, CENPF, MKI67* and *UBE2T.*

### 2.7. Risk of Recurrence Score

The research-based ROR score, which incorporated correlation with the four main subtypes (excluding the normal-like subtype) as well as the proliferation score and tumor size, was calculated as follows [1]: ROR = 54.7690·(−0.0067⋅Basal + 0.4317⋅HER2 − 0.3172⋅LumA 
+ 0.4894⋅LumB + 0.1981⋅prolif + 0.1133⋅T + 0.8826)
where Basal, HER2, LumA, and LumB denote the Pearson correlations to the respective subtypes, T represents the tumor size (T = 1 if tumor size > 2 cm, T = 0 otherwise), and prolif is the proliferation score, as described above. The score ranges from 0 to 100, where a high value means high risk. This continuous score can be categorized into three groups (low, intermediate, and high risk), with individual cutoffs depending on node status. If no nodes are involved, the cutoffs for low, intermediate, and high risk are 40 and 60; when 1–3 nodes are involved, the cutoffs are 15 and 40; and with 4 or more nodes involved, the tumor is automatically assigned to the high-risk category [1]. 

### 2.8. Treatment Recommendation

A potential change in systemic treatment recommendation was independently evaluated by two oncologists, based on the Norwegian national guidelines, including use of the risk of recurrence score [13]. 

## 3. Results

The complete workflow of this study is presented in Figure 1. The output from the approved Prosigna test, based on the macrodissected FFPE tumor tissue, is termed Subtype-Prosigna and ROR-Prosigna; the research-based subtypes and risk scores calculated from the macrodissected FFPE tumor tissue are termed Subtype-Macro and ROR-Macro; and finally, the subtypes and scores calculated based on FF bulk tumor tissue are termed Subtype-Bulk and ROR-Bulk. 

### 3.1. The Research-Based ROR-Macro Recapitulates the Approved ROR-Prosigna in FFPE Tumor Tissue

To assess our ability to classify breast tumors using the raw gene expression counts from the nCounter platform, we compared our calculated research-based subtypes (Subtype-Macro) to the approved subtypes from the Prosigna assay (Figure 2A and Appendix A). There was a high degree of overlap in the subtype classification, with 80/94 (85%) concordant cases between the two approaches. Importantly, for each of the 14 tumors with a mismatch in subtype call, the second-highest subtype in the research-based Subtype-Macro was identical to Subtype-Prosigna (the subtypes called by the Prosigna algorithm) in all but one case (see Appendix A for details). For the discordant cases, the distance between the highest and second-highest correlations was relatively small (marked by black lines in Figure 2A). We also compared our subtyping approach to open-source PAM50 subtyping in the R/Bioconductor package genefu [14], and obtained a high level of concordance (Appendix A). Most importantly, the risk scores ROR-Prosigna and ROR-Macro were highly correlated, with an r^2^ = 0.958 (Figure 2B). Overall, these results show that a research-based approach using the same gene expression counts is able to recapitulate the prognostic score from the Prosigna assay. 

### 3.2. Comparison of ROR Scores Obtained from Macrodissected FFPE and FF Bulk Tumor Tissue 

In a similar approach using the RNA sequencing data from FF bulk tumor tissue, the risk score ROR-Bulk was calculated and compared with ROR-Prosigna (Figure 2C). The results show a wider spread and, thus, a smaller r^2^ = 0.764. We observed the largest differences in the middle range of the ROR scale, which was dominated by luminal tumors. The cause of this variation between the two approaches may be differences in the formulae (research-based versus Prosigna algorithm) and/or variation in the gene expression in the samples (macrodissected FFPE versus FF bulk tissue). As shown in Figure 2B, when only the formulae differed, a small subtype-specific variation was observed, wherein the basal-like tumors obtained slightly higher ROR scores with the research-based approach compared to ROR-Prosigna. This is not observed in Figure 2D, where only the research-based formula is used (Subtype-Macro versus Subtype-Bulk). Therefore, the observed variation between ROR-Bulk and ROR-Prosigna (Figure 2C) must also include differences in gene expression measured in macrodissected FFPE tumor tissue and in FF-bulk tumor tissue. 

To investigate the potential impact of tissue selection and analytical formulae on treatment decisions, we focused on the subgroup of 54 node-negative patients eligible for the Prosigna test in the clinical routine, i.e., the patients with estrogen-receptor-positive (ER+), human epidermal growth factor receptor 2 (HER2)-negative disease. We adopted the clinical thresholds for the Prosigna test and divided the patients into three groups, labeled high, intermediate, or low risk. When comparing the results from ROR-Macro with those from ROR-Bulk for these three groups, 13/54 (24%) tumors were discordant in their risk group classification (Figure 2E; Table 2), with potential implications for adjuvant treatment decisions. The difference in absolute ROR scores in the 13 samples ranged from −25.7 to 44.06 units (with the mean absolute difference being 20.58). Including information from standard histopathological parameters, and using the prevailing national guidelines [13], the difference in ROR score classification would potentially change the systemic treatment recommendation in 6 out of 13 cases if ROR-Bulk were to be used—four escalated treatments and two de-escalated treatments (see Table 2 for more details). In two of these cases, escalation to chemotherapy was indicated. In seven cases with discrepant ROR score categories, adjuvant treatment recommendations remained unchanged. 

To understand the cause of the observed differences between the ROR measurements, we examined the different components of the risk calculations. When correlating the expression from macrodissected FFPE tumor tissue with the expression from FF bulk tumor tissue on a per-gene basis, we observed a high correlation for the majority of genes (Appendix A); all but 4 had a correlation higher than 0.75, including all 18 proliferation-associated genes (marked in orange). Genes coding for cytokeratins were among the least correlated genes when comparing their expression values from macrodissected FFPE tissue with their expression values from FF bulk tumor tissue.

### 3.3. Higher Proportion of the Normal-Like Subtype in Data from FF Bulk Tumor Tissue Impacts ROR Score 

To further study the impact of the gene expression profiles obtained from bulk tissue, we compared the correlations with the centroids from bulk tumor tissue to the Prosigna subtypes (Figure 3A). A substantial proportion of cases in the bulk tumor data showed high correlation with the normal-like centroid, with 13 out of 94 tumors classified as normal-like (Table 3). We obtained similar results when using genefu to estimate the subtypes—11 out of the 94 bulk tumors were classified as normal-like (Appendix A). In contrast, in the macrodissected FFPE tissue-derived data, only one sample was classified as normal-like (this same sample was classified as luminal A by the Prosigna test). We also performed an unsupervised principal component analysis (PCA) on both the FFPE- and FF-tumor-derived data. The normal-like tumors formed a separate cluster clearly separated from the other four subtypes (Appendix A). 

Moreover, when comparing proliferation scores, we found the tumors of the normal-like subtype to have lower cell proliferation when using the data from FF bulk tumor tissue compared to macrodissected FFPE tissue (Figure 3B). This finding seems to be of particular clinical value, since we found the same trend when we compared ROR scores (Figure 3C). This is consistent with tumor and normal breast epithelial cellularity across the subtypes as determined by histopathological evaluation, highlighting the normal-like subtype as a subtype with a low tumor cell percentage (Figure 3D) and a high percentage of normal breast tumor epithelium (Figure 3E). Overall, this shows that the contribution of RNA from non-tumor breast cells has an impact on the ROR scores calculated from bulk tumor gene expression data.

## 4. Discussion

Genomic assays are increasingly used in clinical practice to assess the benefits of adjuvant chemotherapy. The Prosigna gene signature measures the expression of 50 genes in order to determine both the intrinsic subtypes and a risk score in early breast cancer, and is based on digital counts of RNA molecules in an assay optimized for FFPE tissue. In this study, we carried out a risk of recurrence analysis of a unique set of paired FFPE/FF samples from a set of 94 early breast carcinomas. From analysis of FFPE tumor tissue, the clinically approved Prosigna scores were available, serving as an important benchmark against which data from RNA sequencing of FF bulk tissue were compared. The research-based estimates of ROR-Macro were highly correlated with ROR-Prosigna. When comparing the ROR scores obtained from FFPE vs. FF tumor tissue (i.e., ROR-Macro vs. ROR-Bulk), we found discrepancies, but mostly for luminal tumors in the intermediate-risk range. For several patients, the disagreement was distinct, with potential impact on the use of systemic treatment. For the patients in this study, however, several of the discrepant scores would not have influenced treatment decisions, as these also depend on the complete clinicopathological information, as well as the judgement of the treating physician. In only 2 of the 54 node-negative cases would this discrepancy have potentially changed the use of chemotherapy. Although four cases of altered use of endocrine treatment were indicated, we realize that endocrine treatment may be offered to all ER+ breast cancer patients, irrespective of tumor size and biological characteristics. Nevertheless, the observed discrepancy in ROR scores between ROR-Macro and ROR-bulk should be noted; as of today, only macrodissected tissue input has sufficient documentation for clinical utility. 

We observed a higher proportion of the normal-like subtype in the data obtained from FF bulk tissue compared to the FFPE tumor data (13 versus 1). These same tumors were associated with low tumor cellularity, which reduced the ROR estimates. In contrast, the normal-like subtype was practically absent in data obtained from macrodissected FFPE tumor tissue. The clinical implications (change in adjuvant treatment) for the increased proportion of the normal-like subtype for ROR-bulk in this study were probably diminished by a corresponding luminal A categorization by ROR-Macro in the majority of these cases, with endocrine treatment/no treatment decisions predominating in both groups. Due to the similarity to normal breast tissue, it has been debated whether the normal-like subtype is simply a result of RNA contributed by non-tumor cells [15,16]. A previous study showed that normal tissue, concurrently sampled with tumor tissue, is an important source of bias for the PAM50 signature [17]; with in silico contamination of normal tissue, the authors showed that tumors moved from being predicted as more aggressive subtypes to less aggressive subtypes. Nielsen et al. also noted biased subtype calls and a possible effect on ROR score when including non-tumor tissue [18]; however, this was less of a concern when including non-tumor tissue as part of the RNA extraction procedure and subsequent gene expression analysis, as would be the case in clinical practice. In addition, Guedj et al. showed that the rate of non-diploid cells varied among the molecular subtypes as estimated from SNP data, but that this was not a particular feature of the normal-like subtype [19]. As some tumors may have high lymphocyte infiltration, we investigated whether this could explain the discordance in ROR, but the variation in immune cell infiltration assessed by histopathological examination was not the reason for the discrepancy between ROR-Macro and ROR-Bulk (data not shown). Hence, this lends support to our assumption that the variation in prediction is due to tumor heterogeneity in the biopsies, but that the clinical implications are uncertain. 

The limitations of tissue sampling in this study and the lack of repetitive sampling to properly assess reproducibility warrant further investigation. Nevertheless, the variation introduced by sampling different parts of the tumor should be random, and will therefore cause a small bias in both positive and negative directions in the ROR score. Within our analysis, two different technologies were used: RNA sequencing, and multiplexed digital counting. However, focusing specifically on the technical differences, we believe that they should not influence the correlation with the normal-like centroid. Picornell et al. reported a high concordance between the two platforms when comparing subtype calling [20]. Furthermore, Vallon-Christersson et al. compared stratification based on a wide range of gene signatures, using RNA sequencing profiles in a large population-based breast cancer cohort [21]; they found agreement between most signatures in the low- and high-risk groups for the ER+/HER− tumors, and less agreement within the intermediate-risk samples; this is consistent with our results, and stresses that there is a continuous need for improved risk predictors in this patient population. 

## 5. Conclusions

In this study of 94 early-stage breast carcinomas, we observed a general agreement of r^2^ = 0.79 between ROR assessed from macrodissected FFPE tumor tissue and ROR from fresh-frozen bulk tumor tissue. Overall, we observed a higher proportion of the normal-like subtype using RNA extracted from bulk compared to macrodissected tissue. The normal-like tumors showed lower tumor cell content, yielding lower ROR scores, but with limited effect on the treatment decisions. In the subset of ER+HER2-pN0 disease (n = 54), treatment recommendation would potentially have changed for 11% of the patients, including both escalated and de-escalated treatment. Our study highlights the potential of using RNA sequencing from fresh-frozen bulk tissue in diagnostic assays, significantly increasing access to such clinical diagnostic assays. Attention should be paid, however, to tumor cellularity in the context of other clinical parameters that may influence treatment recommendations for particular breast cancer patients. Using FF tissue opens the way for high-quality whole-genome sequencing in clinical practice, including multiple diagnostic signatures and mutational calls, thereby significantly improving diagnostic yield by reflecting the whole molecular portrait of the tumor. As genomic predictors and targeted NGS assays increasingly become available, access to high-quality RNA and DNA provides a wide range of opportunities that may improve decision making in the future.

## Figures and Tables

**Figure 1 cancers-13-06118-f001:**
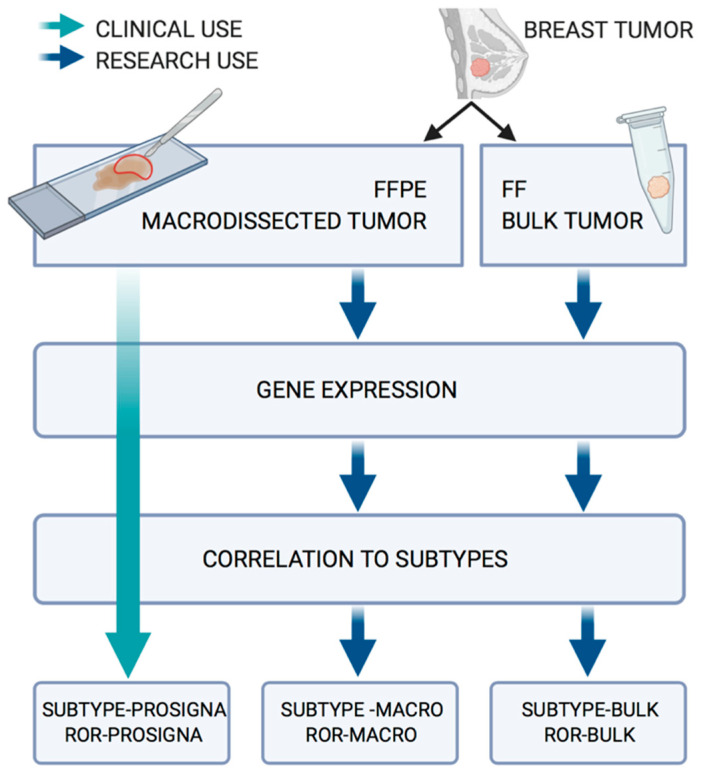
Workflow for the analysis of the 94 paired FFPE/FF samples.

**Figure 2 cancers-13-06118-f002:**
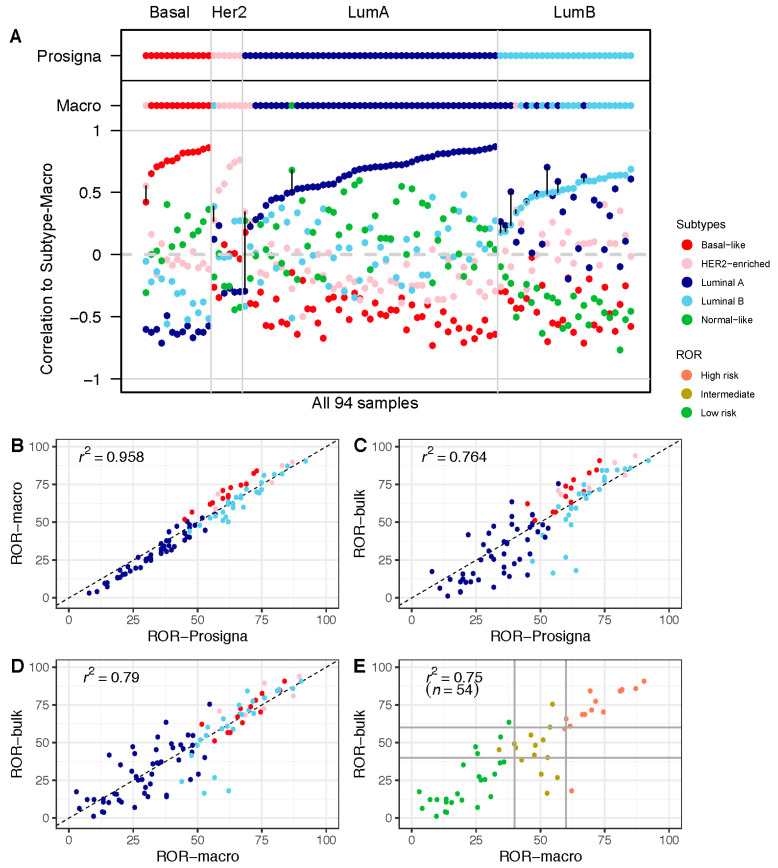
Risk of Recurrence (ROR) in macrodissected FFPE vs. FF bulk tumor tissue: (**A**) Correlation to each of the five PAM50 subtype centroids calculated based on the research-based Subtype-Macro from FFPE tumor tissue. PAM50 subtype calls from Subtype-Prosigna and Subtype-Macro are shown on top for comparison. Correlation coefficients on the *y*-axis, and the 94 breast tumor samples along the *x*-axis, are sorted by increasing correlation with Subtype-Prosigna. Discordant cases between Subtype-Prosigna and Subtype-Macro are marked with black vertical lines. Samples are colored according to subtype (red = basal-like; pink = HER2-enriched; dark blue = luminal A; light blue = luminal B; green = normal-like). (**B**) ROR-Prosigna plotted against ROR-Macro for all 94 patients. (**C**) ROR-Prosigna plotted against ROR-Bulk for all 94 patients. (**D**) ROR-Macro plotted against ROR-Bulk for all 94 patients. (**E**) ROR-Macro plotted against ROR-Bulk for the subgroup of 54 node-negative, ER+, and HER2− breast tumors eligible for the Prosigna test. In panels (**B**–**D**), samples are color-coded by subtype according to Subtype-Prosigna, and in panel (**E**) by ROR-Prosigna (red = high risk; brown = intermediate risk; green = low risk). Horizontal and vertical lines indicate the clinically used thresholds for ROR.

**Figure 3 cancers-13-06118-f003:**
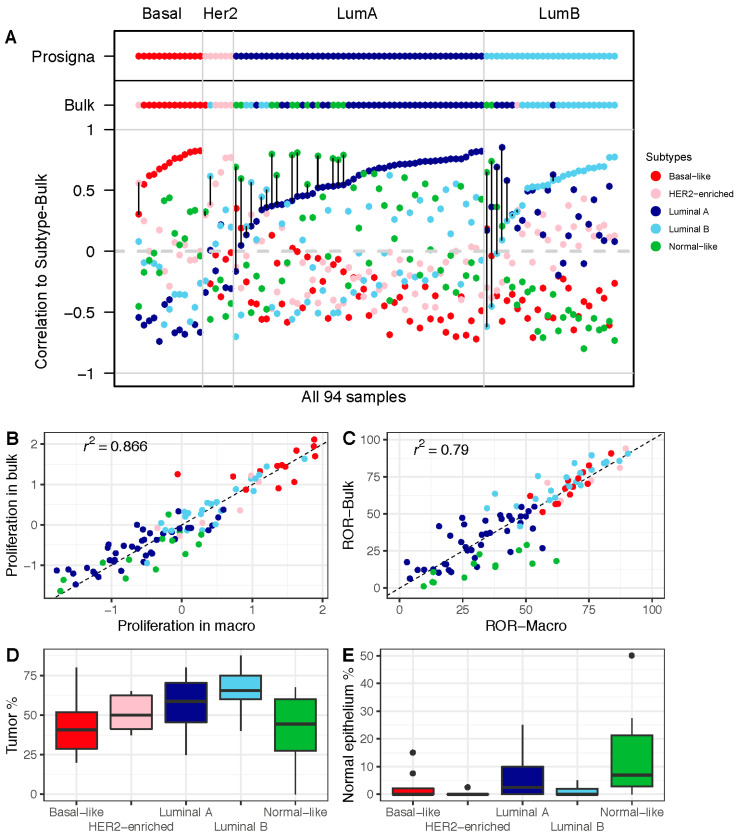
Impact of gene expression in FF bulk tumor tissue on subtype calls and ROR: (**A**) Correlation with each of the five PAM50 subtype centroids, calculated based on the research-based Subtype-Bulk from FF tumor tissue. PAM50 calls from Subtype-Prosigna and Subtype-Bulk are shown on top for comparison. Correlation coefficients on the *y*-axis, and the 94 breast tumor samples along the *x*-axis, are sorted by increasing correlation with Subtype-Prosigna. Discordant cases between Subtype-Prosigna and Subtype-Bulk are marked with black vertical lines. Samples are colored according to subtype (red = basal-like; pink = HER2-enriched; dark blue = luminal A; light blue = luminal B; green = normal-like). (**B**) Proliferation score from macrodissected FFPE plotted against FF bulk tumor tissue. (**C**) ROR-Macro plotted against ROR-Bulk. (**D**) Boxplots showing estimated tumor content across subtypes. (**E**) Boxplots showing estimated normal breast epithelium across subtypes. In panels (**B**–**E**), all 94 samples are color-coded by subtype according to Subtype-Bulk.

**Table 1 cancers-13-06118-t001:** Clinical overview of all 94 samples, and the subset of 54 that were ER+HER2-pN0.

All Patients	94 (%)	54 (%)
Prosigna subtype		
Basal-like	13 (14%)	1 (2%)
HER2-enriched	6 (6%)	
Luminal A	49 (52%)	34 (63%)
Luminal B	26 (28%)	19 (35%)
T status		
T1b	12 (13%)	9 (17%)
T1c	46 (49%)	28 (52%)
T2	32 (34%)	15 (28%)
T3	3 (3%)	2 (4%)
T4	1 (1%)	
N status		
pN0	64 (68%)	54 (100%)
pN1	25 (27%)	
pN2	5 (5%)	
Histological grade		
I	17 (18%)	12 (22%)
II	46 (49%)	31 (57%)
III	31 (33%)	11 (20%)
HER2 status		
Positive	6 (6%)	
Negative	82 (87%)	50 (93%)
Missing	6 (6%)	4 (7%) *
Ki67		
< 15%	12 (13%)	11 (20%)
15–30%	25 (27%)	16 (30%)
≥ 30%	56 (60%)	26 (48%)
Missing	1 (1%)	1 (2%)
Histological subtype		
Ductal	61 (65%)	34 (63%)
Lobular	12 (13%)	9 (17%)
Other	21 (22%)	11 (20%)

T = Tumor; N = Node; * The four samples with missing information on HER2 IHC showed negative status on FISH in three samples (BC-13, 35, 41), and low-level amplification in one case (BC-14).

**Table 2 cancers-13-06118-t002:** ROR scores and corresponding categories for the 13 discordant cases between ROR-Macro and ROR-Bulk, ordered in rows according to the difference in risk score. Potential change of treatment recommendations within this ER+ HER2+ subpopulation is indicated in the systemic treatment recommendation column. Treatment is based on the Norwegian guidelines, including the use of the Prosigna test [13].

Sample ID	Subtype	Prosigna	Prosigna (Cat)	ROR- Macro (Cont.)	ROR- Macro (Cat)	ROR- Bulk (Cont.)	ROR- Bulk (Cat)	Systemic Treatment Recommendation; Macro → Bulk	pT	Grade	Ki67	Histological Subtype
BC-34	Luminal A	39	Low	37.87	Low	63.56	High	No adjuvant → Endo	T1b	II	>=30%	Ductal
BC-30	Luminal A	32	Low	24.9	Low	47.3	Inter	No adjuvant → Endo	T1c	I	15–30%	Ductal
BC-35	Luminal A	57	Inter	54.86	Inter	75.53	High	Endo → Chemo	T2	II	>=30%	Ductal
BC-20	Luminal A	39	Low	34.69	Low	53.77	Inter	No change	T1c	II	Missing	Ductal
BC-85	Luminal A	30	Low	25.65	Low	42.83	Inter	No change	T1b	II	15–30%	Ductal
BC-72	Luminal A	41	Inter	34.1	Low	45.41	Inter	No change	T2	I	15–30%	Ductal
BC-17	Luminal A	47	Inter	39.94	Low	49.07	Inter	No change	T2	II	>=30%	Ductal
BC-88	Luminal B	57	Inter	53.93	Inter	60.06	High	Endo → Chemo	T1c	II	15–30%	Ductal
BC-23	Luminal A	45	Inter	42.71	Inter	38.43	Low	Endo → no adjuvant	T1c	I	15–30%	Ductal
BC-58	Luminal A	46	Inter	50.42	Inter	28.99	Low	No change	T1c	II	>=30%	Ductal
BC-47	Luminal B	60	Inter	56.66	Inter	26.85	Low	No change	T1c	II	>=30%	Lobular
BC-38	Luminal B	55	Inter	52.69	Inter	16.35	Low	No change	T1c	II	>=30%	Ductal
BC-70	Luminal B	64	High	62.18	High	18.12	Low	Endo → no adjuvant	T1b	II	>=30%	Ductal

**Table 3 cancers-13-06118-t003:** Comparison of subtype calls using Subtype-Bulk and Subtype-Macro: subtypes based on gene expression obtained from fresh-frozen bulk tumor biopsies are shown in columns, and subtypes based on gene expression obtained from macrodissected FFPE sections are shown in rows.

Subtype Macro	Subtype-Bulk				
	Basal-Like	HER2-Enriched	Luminal A	Luminal B	Normal-Like
Basal-like	12				
HER2-enriched	1	6			2
Luminal A			39	6	9
Luminal B				17	1
Normal-like					1

## Data Availability

R codes are available as Appendix A. Normalized log2-transformed nCounter counts for the PAM50 genes can be found in Appendix A. Normalized RNA-Seq data from the PAM50 genes can be found in Appendix A. The raw genome-wide RNA-Seq data are available from EGA (accession number EGAS00001003631), and normalized genome-wide log2-transformed FPKM data are available from GEO (accession number GSE135298).

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
