# Peer review of "Sample Preparation Approach Influences PAM50 Risk of Recurrence Score in Early Breast Cancer"

_cancers, 2021, doi:10.3390/cancers13236118_

Round 1

Reviewer 1 Report

An interesting study analyzing how the use of formalin-fixed paraffin-embedded  tissue sections or a fresh frozen  bulk tissue without macrodissection may influence PAM50 gene expression subtypes and the associated risk of recurrence, showing similar results with the two methods in a sample of 94 patients; only minor queries:

line 56 you should add: ", as a more complete profile may guide the clinician into the correct diagnosis" and cite an article such as doi: 10.1007/s40264-021-01071-1.

Conclusions should be expanded better highlitìghting the potential use of RNA sequencing from fresh frozen bulk tissue.

Author Response

Response to Reviewer 1 Comments

We would like to thank the reviewer for the suggestions to improve our manuscript. Below are our responses to each point raised.

An interesting study analyzing how the use of formalin-fixed paraffin-embedded  tissue sections or a fresh frozen  bulk tissue without macrodissection may influence PAM50 gene expression subtypes and the associated risk of recurrence, showing similar results with the two methods in a sample of 94 patients; only minor queries:

Point 1:  -line 56 you should add: ", as a more complete profile may guide the clinician into the correct diagnosis" and cite an article such as doi: 10.1007/s40264-021-01071-1.

Response 1: The reviewer suggests that we add a sentence in line 56-57 in the beginning of the Introduction. However, we feel that the next sentence in our current manuscript in line 57-58: " The score refines patient stratification for improved treatment decisions, leading to alleviation of chemotherapy for some patients” and the four references to previous articles that demonstrate the utility of the Prosigna score /PAM50 gene signature, are clearly stating the relevance of the score for clinical use. We interpret the suggested reference “Emerging Skin Toxicities in Patients with Breast Cancer Treated with New Cyclin-Dependent Kinase 4/6 Inhibitors: A Systematic Review” as not suitable for this particular use.

Point 2:  Conclusions should be expanded better highlighting the potential use of RNA sequencing from fresh frozen bulk tissue.

Response 2: The reviewer suggests that we expand our conclusions to better highlight the potential of RNA sequencing. We have changed the last part of the Discussion slightly and expanded the conclusions to emphasize the potential for using sequencing from bulk tumor tissue in a clinical diagnostic setting.

Reviewer 2 Report

Manuscript "Sample preparation approach influences PAM50 risk of recurrence score in early breast cancer" is an interesting scientific study. It also has a significant practical aspect enabling the diagnosis of women with breast cancer.
The manuscript development is correct. After a short introduction, the Authors describe in detail the methods used and the results obtained. The graphic representation for the analysis of the 94 paired FFPE / FF samples deserves a special mention.
Abstract and conclusions are properly elaborated.
Due to the possibility of practical use of the research methods described by the Authors, the manuscript "Sample preparation approach influences PAM50 risk of recurrence score in early breast cancer" should be accepted for publication in CANCERS and printed in the present form.

Author Response

Response to Reviewer 2 Comments

Manuscript "Sample preparation approach influences PAM50 risk of recurrence score in early breast cancer" is an interesting scientific study. It also has a significant practical aspect enabling the diagnosis of women with breast cancer.
The manuscript development is correct. After a short introduction, the Authors describe in detail the methods used and the results obtained. The graphic representation for the analysis of the 94 paired FFPE / FF samples deserves a special mention.
Abstract and conclusions are properly elaborated.
Due to the possibility of practical use of the research methods described by the Authors, the manuscript "Sample preparation approach influences PAM50 risk of recurrence score in early breast cancer" should be accepted for publication in CANCERS and printed in the present form.

Response: We thank the reviewer for the positive comments about our manuscript.

Reviewer 3 Report

This manuscript describes the variations in the evaluation of the risk of recurrence score with two sample preparation procedures. The study provides a critical view on the use of PAM50-based prediction, which is derived from the microdissected or bulk frozen samples. While the viewpoint is important, the interpretation and conclusion can be improved by conducting additional experiments.

Major

  • variations: the prediction can vary based on three major reasons: the variations in biological samples, technical procedures, and measurement errors. The authors claim that the observed variations are due to the technical procedures that alter the biological samples. The interpretation is reasonable when the other variations are removed and the prediction is reproducible. The prediction based on RNA analysis in this study with microdissected or bulk frozen samples was conducted once and no reproducibility is shown. This procedure is not appropriate, and the authors need to show reproducibility.
  • validation: the authors interpreted the potential reason for the variation with the frozen samples. It is recommended to evaluate and validate the interpretation by conducting the validation-targeted experiment. If the mixture of non-tumor cells is a potential cause, the authors should be able to validate their interpretation by adjusting the starting materials for RNA analysis.
  • Concluding remark: It is recommended that the authors provide a clear recommendation of the use of PAM50.

Minor

  • PAM50 genes: It is recommended to include a description of PAM50 genes in the introduction.
  • Materials and methods: The current description is not sufficiently detailed. It is recommended to make a few subsections and provide in-depth descriptions.

Author Response

Response to Reviewer 3 Comments

We thank the reviewer for the comments and suggestions to improve our manuscript. Below are our responses.

This manuscript describes the variations in the evaluation of the risk of recurrence score with two sample preparation procedures. The study provides a critical view on the use of PAM50-based prediction, which is derived from the microdissected or bulk frozen samples. While the viewpoint is important, the interpretation and conclusion can be improved by conducting additional experiments.

Major

Point 1:  variations: the prediction can vary based on three major reasons: the variations in biological samples, technical procedures, and measurement errors. The authors claim that the observed variations are due to the technical procedures that alter the biological samples. The interpretation is reasonable when the other variations are removed and the prediction is reproducible. The prediction based on RNA analysis in this study with microdissected or bulk frozen samples was conducted once and no reproducibility is shown. This procedure is not appropriate, and the authors need to show reproducibility.

Response 1: Thanks for pointing to this important aspect of reproducibility in prediction. This study was not designed to assess the influence of variations on prediction; it was rather taking the advantage of two sets of RNA measurements obtained from the same tumors but with different processing protocols and investigate the concordance in subtype calls and the potential clinical implications of using RNA sequence-based gene expression measurements. To demonstrate reproducibility in prediction in studies like this is extremely challenging due to limited availability of patient material. Here we have a good study size (n=94) yielding a substantial number of cases per subtype to attempt to compensate for the challenge of reproducibility across repetitive samples. We also refer to Nielsen et al. (ref 18 in our manuscript) and Picornell (ref 19 in our manuscript) for analytical validation of Prosigna and for concordance between the two different gene expression platforms. We have emphasized this limitation in the last part of the Discussion.

Point 2: validation: the authors interpreted the potential reason for the variation with the frozen samples. It is recommended to evaluate and validate the interpretation by conducting the validation-targeted experiment. If the mixture of non-tumor cells is a potential cause, the authors should be able to validate their interpretation by adjusting the starting materials for RNA analysis.

Concluding remark: It is recommended that the authors provide a clear recommendation of the use of PAM50.

Response 2: This is also a valid point and an excellent approach to evaluate the reason for the variations observed. However, in this study this is not possible as we used RNA extracted from bulk frozen tissue from the same patients as was used for the study with the Prosigna assay. This particular issue was addressed in the study by Nielsen et al. (ref 18), who found that the amount of tumor-adjacent tissue had an effect on subtype determination and that this was eliminated by the macrodissection step in the Prosigna protocol. Further Elloumi and colleagues (ref. 17 in our manuscript) simulated normal cell contamination of the tumor analyses they performed and concluded that the amount of normal cells in tumors has effect on genomic predictors. Finally, Guedj et al., (now reference 20 in our manuscript) used SNP data to estimate the amount of non-tumor cells in the breast tumor samples analyzed in their study and found significant differences among the molecular subgroups. These studies support our assumption that heterogeneity in the frozen tumor tissue including varying normal cell contamination contribute to the variation observed.

With respect to recommending the use of PAM50; in the ESMO guidelines, Prosigna/PAM50 is recommended to gain additional prognostic and/or predictive information to complement pathology assessment and to predict the benefit of adjuvant chemotherapy. Prosigna is as of today, the only PAM50-based gene expression test has clinical utility. However, we believe that a PAM50 gene expression assay based on RNA sequencing data can be implemented in clinical use provided it is sufficiently tested in clinical studies. We have changed the last part of the discussion and expanded the conclusions to better reflect this. See also response to Reviewer 1.   

Minor

Point 3: PAM50 genes: It is recommended to include a description of PAM50 genes in the introduction.

Response 4: We have emphasized that the PAM50 signature consists of 50 genes and added references to Perou et al. 2000 and Parker et al. 2009 who first described the intrinsic genes and subsequently arrived at the PAM50 gene set in lines 59-60. Since then, these 50 genes have been included in numerous articles and we feel that it will negatively impact the readability of the text if we list all 50 genes in the Introduction.

Point 4: Materials and methods: The current description is not sufficiently detailed. It is recommended to make a few subsections and provide in-depth descriptions.

Response 5: We have expanded the Materials and Methods, included subsections and provided some more details where we found appropriate.

Round 2

Reviewer 1 Report

The paper improved after revisions 

It is eligible to be published

Reviewer 3 Report

The responses by the authors are adequate. Thank you.